# Early-onset burdensome multimorbidity: an exploratory analysis of sentinel conditions, condition accrual sequence and duration of three long-term conditions using the 1970 British Cohort Study

Sebastian Stannard [1], Emilia Holland,[1] Sarah R Crozier,[2,3] Rebecca Hoyle,[4] Michael Boniface,[5] Mazen Ahmed,[5] James McMahon,[3] William Ware,[3] Zlatko Zlatev,[5] Nisreen A Alwan,[1,3,6] Simon DS Fraser [1]

For numbered affiliations see end of article.

**Correspondence to**
Dr Sebastian Stannard;
S.J.Stannard@soton.ac.uk

## ABSTRACT

**Objectives** The prevalence of multiple long-term condition (LTC) multimorbidity is increasing with younger onset among socioeconomically deprived populations. Research on life course trajectories towards multimorbidity is limited and early-onset multimorbidity poorly characterised. Understanding sentinel conditions (the first LTC occurring in the life course), the sequence of LTC accrual and the permanency of the reporting of LTCs may help identify time points for prevention efforts. We used a longitudinal birth cohort to estimate the prevalence of a common three-condition early-onset multimorbidity (multiple long-term condition multimorbidity (MLTC-M)) group at midlife, describe the frequency of sentinel conditions, the sequence of LTC accrual and explore the permanency of one of these conditions: psychological distress.

**Setting** 1970 British Cohort Study (BCS70).

**Participants** 17 196 cohort members born in 1970.

**Outcome measures** Prevalence of the most common three-condition multimorbidity group at age 46. The nature and timing of sentinel conditions, the sequencing patterns of subsequent LTC accrual and the permanency of the reporting of psychological distress.

**Results** At age 46 high blood pressure, psychological distress and back pain were the most common three-condition MLTC-M group, (4.3%, n=370). A subgroup of 164 (44.3%) people provided complete information on LTC across all time points. Psychological distress measured by the Malaise Index was the most common sentinel condition, occurring in 25.0% (n=41), followed by back pain (22%, n=36). At age 26, 45.1% (75/164) reported their sentinel condition. The most common sequence of LTC accrual was the co-reporting of psychological distress and back pain followed by high blood pressure. Almost one-third (30.5%, n=50) reported a variation of psychological distress across the adult life course.

**Conclusion** In these exploratory analyses, psychological distress and back pain were the most common sentinel conditions, and along with high blood pressure these three

conditions represented the most common three-condition MLTC-M group. These analyses suggest that birth cohorts, like the BCS70, may usefully inform life course-multimorbidity research.

## STRENGTHS AND LIMITATIONS OF THIS STUDY

⇒ The use of a prospective longitudinal approach provided the opportunity to analyse the time-sequenced nature of the data.

⇒ Knowing the temporal sequencing of long-term conditions enabled us to describe the frequency of sentinel conditions, the sequence of long-term condition accrual and explore the permanency of the reporting of a long-term condition.

⇒ Using birth cohort data, with repeated validated measures, allowed us to explore and characterise sequences that incorporated the concept of 'burdensome' multimorbidity by combining both mental and physical conditions.

⇒ The study was limited by sample size, wave non-response and no agreed definition of long-term condition.

⇒ The self-reporting of long-term conditions and the lack of ethnic diversity resulted in the probable under-reporting of multimorbidity.

## INTRODUCTION

In the UK, the prevalence of multiple long-term condition multimorbidity (MLTC-M) has increased over the last two decades and is projected to continue to rise.[1] This presents a significant challenge to health and prevention services.[2]

MLTC-M occurs earlier in the life course among people from more socioeconomically disadvantaged backgrounds.[3] While often considered a condition of older age, the

BMJ

majority of people living with multimorbidity are under 65 years of age.[3] Earlier and greater accrual of long-term conditions (LTCs) may be influenced by the timing and nature of the exposure to key risk factors or other LTCs and the influence of wider determinants at different life stages.

Research on MLTC-M occurring by midlife remains limited.[4] Growing evidence suggests that the sequence of accrual of LTCs varies considerably and influences outcomes.[5] The sentinel condition (the first LTC occurring in the life course in the development of MLTC-M), may play an important role. Sentinel conditions may significantly influence subsequent behaviour and other factors that influence the eventual cluster of LTCs, in turn influencing the nature and risk of important outcomes. Diagnosis of a condition is associated with actions such as medication prescription, specialist referral and self-management advice, that may alter future trajectories of risk. For example, hypertension diagnosis leading to antihypertensive prescription and physical activity advice reducing future stroke risk. The time point of diagnosis therefore becomes an important determinant of future events.[6]

The sequence of LTC development may also be a key determinant of the nature of eventual LTC clusters.[7] In a longitudinal analysis using UK primary care data from London, Ashworth et al identified that multimorbidity in those under 65 tended to start with depression or serious mental illness followed by combinations of obesity, chronic pain and diabetes.[7] In contrast, for those aged 65 and over, diabetes and coronary heart disease were the predominant sentinel conditions, followed by combinations of chronic kidney disease, heart failure and stroke.[7] Additionally, the prevalence of multimorbidity tends to rise steeply in midlife,[8] and middle-aged people with multimorbidity tend to have poorer health-related quality of life.[9]

Across the life course, both physical and mental LTCs may not be permanent and may be of variable severity. For example, mental health conditions such as anxiety and depression are often not experienced as a 'fixed state'.[10–12] However, those with a persistent and chronic trajectory of depression tend to have poorer outcomes including suicide, a decrease in physical activity, ischaemic heart disease and mortality.[13–16] Using birth cohort data, with repeated validated measures, affords the opportunity to explore mental health trajectories across the life course. Understanding the patterns of the reporting of mental health for those with MLTC-M may be important for identifying high-risk temporal life periods in the development of multimorbidity.

There also remains limited research on the differential burden experienced by patients with different multimorbidity clusters. Despite research suggesting the relationship between mental and physical health conditions is bidirectional,[17] there remains a need to understand what constitutes burden and complexity for people with multimorbidity. In particular, the National Institute for Health and Care Excellence multimorbidity guidelines recommend establishing better understanding of disease burden encompassing both complex disease and combined mental and physical health.[18]

Previous studies have used limited time frames and limited analyses of the sequence of accrual of LTCs.[4] Using birth cohort data to study multimorbidity at midlife is underexplored. Gondek et al used data from the 1970 British Cohort Study (BCS70) suggesting that multimorbidity (two LTCs) is common at age 46–48 (34%).[19] Building on this work, we aimed to study the most common three-condition multimorbidity group at age 46 using the BCS70 in order to explore and characterise sequences and incorporate the concept of 'burdensome' multimorbidity by combining both mental and physical conditions. We aimed to describe the nature, timing and recurrence of the sentinel condition for those reporting the most common three-condition MLTC-M, to illustrate the sequence of accrual of LTCs following the sentinel condition and describe the permanency of the reporting of psychological distress.

## METHODS

### Birth cohort study

The BCS70 followed 17 196 participants born in 1970 from across England, Scotland and Wales. Since birth, data was collected in 'sweeps' roughly every 4 years resulting in 10 sweeps of data collection to date. The BCS70 data sets generated and analysed in the study are available in the UK Data Service repository: BCS70,[20] accessed via an End User License Agreement. Extensive information was collected on topics including socioeconomic circumstances, family background, cognitive development, education, employment, partnerships, fertility, health and health-related behaviour.[21] Data was collected using a range of survey methods including self-reported and measured outcomes. A user guide for the data set can be found at the Centre for Longitudinal Studies.[22] A cohort profile provides information about the background to the study and full survey methods for each sweep.[21 23]

### Study sample

Information was collected on 17 196 babies born in 1970. At age 46, 8581 (49.9%) of the original population at birth responded while 8536 of them provided information on LTCs. To derive the sentinel condition and accrual sequence we considered the MLTC-M group at age 46 who provided information across all the sweeps of data collection in adulthood (age 26, 30, 34, 38, 42 and 46) (n=164/370).

### Patient and public involvement statement

Patient and public involvement and engagement (PPIE) co-investigators have been involved at every stage of the research process. Their lived experiences were important for informing the design of the research, incorporating the views of others and shaping our ideas. In particular,

PPIE input directed us to consider multimorbidity clusters that were both burdensome and complex and to consider the permanency of the reporting of psychological distress.

## Variables

### Socio-demographic measures at age 46

A number of area-level and individual-level socio-demographic variables were recorded at age 46. Measures included sex (men/women), the Index of Multiple Deprivation (IMD) (in quintiles) and highest academic qualification (no qualification, GCSE or equivalent, A levels or equivalent or degree and higher). Other measures included a three-measure index of housing tenure (own/mortgage, social rent and private rent/other), measure of local authority or housing association housing (yes/no), employment status (employed, unemployed, sickness/disability or caring for family) and a measure of receipt of benefits (yes/no).

### Selecting MLTC-M outcome at age 46

We selected psychological distress, back pain and high blood pressure as our key MLTC-M group at age 46 because our scoping work had suggested that this combination of three LTCs was among the most common MLTC-M group.[24] The other nine three-condition MLTC-M groups encompassed the most common LTCs at age 46, including psychological distress, back pain, high blood pressure, asthma and arthritis. Psychological distress measured via the Malaise Index is a term developed by the UK Data Service,[25] and used widely within the context of the British birth cohort studies. Although, psychological distress is not per se pathological and therefore not strictly morbidity we use it as an indicator of mental health. The common three-condition MLTC-M group selected also explored the concept of 'burdensome' multimorbidity by combining both mental and physical conditions.

### LTCs at ages 26, 30, 34, 38 and 42

To explore the sentinel condition, sequence of accrual and the permanency of the reporting of psychological distress we required information on LTCs across all of the sweeps in adulthood (age 26, 30, 34, 38 and 42). All measures collected between ages 26 and 42 were self-reported and we had no measured blood pressure. If two or more LTCs were recorded at the same sweep we were unable to identify which condition occurred first.

*Psychological distress and anxiety or depression at age 46* was classified if a cohort member had measured high on the Malaise Index,[25] or if they had visited a doctor or specialist for anxiety and/or depression in the last 4 years or since their last interview. The Malaise Index was selected because it was consistently present at sweeps in adulthood. The Malaise Index was self-reported and designed to capture general signs of psychological distress in adulthood including depression and anxiety symptoms.[26 27] Scores above the cut-off (4 or higher) represent

a clinical diagnosis of common mental disorder.[26] Scalar invariance of the measure was found within and across two birth cohort studies, as well as between genders.[28 29] The Malaise Index has good psychometric properties,[30] has a good level of internal consistency and its validity is robust across different populations including the general population and high-risk groups.[31 32]

*Psychological distress at ages 26–42* was calculated using the cross-sectional reporting of the Malaise Index attained at ages 26, 30, 34 and 42, with scores above the cut-off (4 or higher) representing a clinical diagnosis of a common mental disorder.[26]

*Back pain at age 46* was self-reported and a cohort member was classified as having back pain if they reported doctor's diagnosed back pain (cohort member reporting if they had visited the doctor for an issue relating to back pain in the past 4 years or since their last interview) and/or moderate or severe pain (pain severity in the past 4 weeks measured on a 5-point Likert scale). Those reporting moderate or severe pain were assumed to have a condition relating to pain.

*Back pain at ages 26–42* was specified using the cross-sectional self-reporting of back pain at ages 26, 30, 34, 38 and 42. We assumed back pain if the cohort member visited a doctor for 'back pain' or 'backache'. At age 42 we additionally included a measure of pain severity, those reporting moderate or severe pain were assumed to have a condition relating to pain. Age 42 was the first sweep at which pain severity was reported.

*Blood pressure at age 46* was classified if the cohort member had measured hypertension or if they had received a doctor's diagnosis of high blood pressure, since diagnosis of high blood pressure may result in a prescription for antihypertensive medication, thus lowering blood pressure readings. Blood pressure was measured in millimetres of mercury and was used to classify individuals as either having hypertension defined as systolic/diastolic blood pressure of at least 140/90 mm Hg, or not.[33]

*High blood pressure at ages 30–42* was calculated using the cross-sectional self-reporting of high blood pressure. High blood pressure was not recorded at age 26, although we do not feel this will alter our results as the number reporting high blood pressure by age 26 was likely to be extremely small—less than 1%.[34 35] At age 30 the cohort member reported if they had ever had high blood pressure and at ages 34, 38 and 42 the cohort member reported if they visited the doctors for a health problem relating to high blood pressure in the last 4 years or since their last interview.

## Statistical analysis

Descriptive analyses explored the percentage distribution of socio-demographic characteristics for the most common three-condition MLTC-M group compared with the remainder of the cohort being considered in this paper. To compare the differences between groups Mann-Whitney U tests were used.

Prevalence was calculated by dividing the number of cohort member in the MLTC-M group by the number of cohort members in the BCS70 cohort at age 46. The acquisition sequence of LTCs was established by searching each sweep for onset of each LTC. We tabulated sentinel condition, the number of people reporting a sentinel condition at each sweep and the length of time to accrue all three LTCs after reporting the sentinel condition. The sequence of acquisition of psychological distress, high blood pressure and back pain was established by coding the sweep each LTC was first reported and grouping these into patterns dependent on the order of acquisition of LTCs. The permanency of the reporting of psychological distress was established by deriving the reporting of psychological distress (yes/no) at five sweeps (ages 26, 30, 34, 42 and 46).

We displayed findings using a bubble diagram, bar chart and a Sankey diagram. These were constructed using Excel and using online software package 'jsfiddle'.[36] Analysis of the acquisition of sentinel conditions, the sequence of LTCs and the evaluation of the permanency of the report of psychological distress were conducted using the statistical software package Stata IC V.16,[37] and Excel. Mann-Whitney U tests were performed using Stata IC V.16.

## RESULTS
### Multimorbidity cohort characteristics
Psychological distress was the most common single LTC (33.2%, n=2835) (table 1). Back pain and high blood pressure were reported by 32.4% (n=2767) and 23.5% (n=2010), respectively. The most common three-condition MLTC-M was psychological distress, high blood pressure and back pain reported by 4.3% (n=370) (table 1) of whom 164/370 provided additional information on LTCs at ages 26, 30, 34, 38 and 42 (figure 1).

### Socio-demographic characteristics at age 46
At age 46, for those in the most common three-condition MLTC-M group, a larger proportion of participants (24.3%) lived in areas in the lowest IMD quintile compared with the remainder of the BCS70 sample (12.7%) (p<0.001) (table 2). Participants in the MLTC-M group were more likely to report no academic qualifications (38.4% vs 27.7%, p<0.001) and were more likely to be living in social rented accommodation (27.3% vs 9.9%, p<0.001) compared with the remainder of the BCS70 sample. They were also more likely to receive benefits (54.0% vs 36.7%, p<0.001) and were more likely to be off work for issues relating to sickness (19.5% vs 3.5%, p<0.001) compared with the remainder of the BCS70 sample.

### Sentinel conditions for the MLTC-M subgroup (n=164)
For the 164 participants with the most common three-condition MLTC-M at age 46 who also provided information on LTCs at ages 26, 30, 34, 38 and 42, psychological

**Table 1** The frequency of the reporting of long-term conditions at age 46 for the full BCS70 cohort

| | | N (%) |
|---|---|---|
| One LTC | Psychological distress | 2835 (33.2) |
| | Back pain | 2767 (32.4) |
| | High blood pressure | 2010 (23.5) |
| | Asthma | 1003 (11.7) |
| | Arthritis | 652 (7.6) |
| Three LTCs | Psychological distress, back pain and high blood pressure | 370 (4.3) |
| | Arthritis, psychological distress and back pain | 240 (2.8) |
| | Asthma, psychological distress and back pain | 236 (2.8) |
| | Asthma, high blood pressure and psychological distress | 145 (1.7) |
| | Asthma, high blood pressure and back pain | 134 (1.6) |
| | Arthritis, high blood pressure and back pain | 131 (1.5) |
| | Arthritis, high blood pressure and psychological distress | 113 (1.3) |
| | Arthritis, asthma and back pain | 89 (1.0) |
| | Arthritis, asthma and psychological distress | 84 (1.0) |
| | Arthritis, asthma and high blood pressure | 50 (0.6) |
| | Total | 8536 |

BCS70, 1970 British Cohort Study; LTC, long-term condition.

distress (25.0%, n=41), back pain (21.3%, n=35) and the co-reporting of psychological distress and back pain (21.3%, n=35) were the three most common sentinel conditions/combinations (figure 2). Hypertension was the sentinel condition for 9.1% (n=15).

At age 26, 45.1% of the subgroup (n=74/164), reported the sentinel condition (figure 2). Most common sentinel conditions at age 26 included back pain, reported by 15.9% (n=26), the co-reporting of psychological distress and back pain reported by 15.2% (n=25) and the reporting of psychological distress by 12.2% (n=20).

### Sequence of accrual of LTCs for the MLTC-M subgroup (n=164)
For the MLTC-M subgroup (n=164) the most common sequence of LTCs was the co-reporting of psychological distress and back pain followed by high blood pressure (21.3%, n=35) (figure 3). Other common sequences included psychological distress followed by back pain and then high blood pressure (9.8%, n=16) and back pain

followed by high blood pressure and then psychological distress (9.1%, n=15).

## The variation in the reporting of psychological distress for the MLTC-M subgroup (n=164)

For the MLTC-M subgroup (n=164), 16.4% (n=27) reported psychological distress at age 46 only. Psychological distress at all sweeps was reported by 8.5% (n=14), while 18.9% (n=31) reported psychological distress at one sweep and then continued to report psychological distress at all subsequent sweeps and 25.6% (n=42) reported psychological distress at one sweep only and then at age 46 (figure 4). One fluctuation was reported by 21.3% (n=35) in the reporting of psychological distress and this group reported psychological distress at one sweep, then did not report psychological distress at a following sweep and then reported psychological distress again at all further sweeps (no-yes-no-yes/yes-no-yes). Multiple fluctuations were reported by 9.2% (n=15) in the reporting of psychological distress and this group reported psychological distress at one sweep, then did not report psychological distress at a following sweep, then reported psychological distress again at the next sweep and then did not report psychological distress at a following sweep, and finally reported psychological distress at age 46 (no-yes-no-yes-no-yes/yes-no-yes-no-yes) (figure 4). Finally, 17.7% (n=29) reported a U shape psychological distress trajectory with elevated levels of psychological distress in their mid-20s and at midlife.

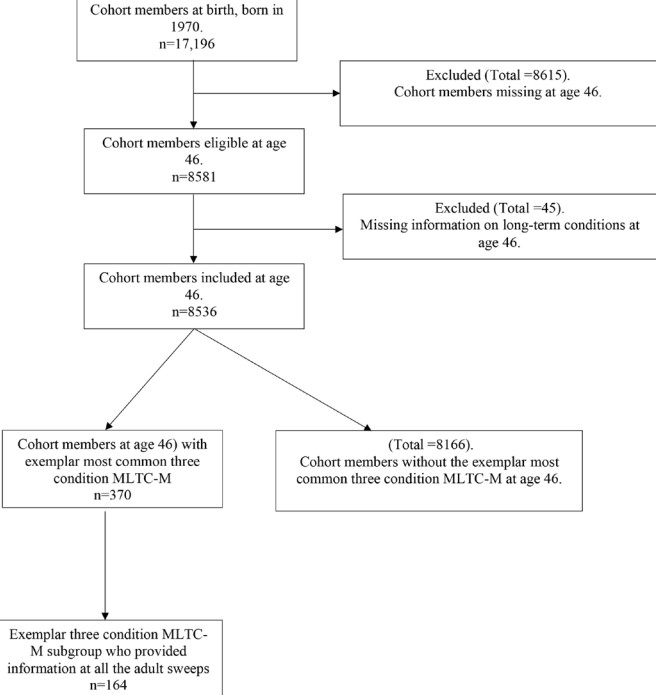

**Figure 1** Selection flow of multiple long-term condition multimorbidity (MLTC-M) sample at age 46.

## DISCUSSION
### Principal findings
In this national birth cohort, 8536 individuals provided information on LTCs at age 46. The most common three-condition MLTC-M grouping—psychological distress, back pain and high blood pressure—was found in 4.3% (n=370) of the people. Descriptive analysis and Mann-Whitney U tests suggests that those with the three-condition MLTC-M were characterised by worse socio-demographic factors. For the 44.3% (n=164) MLTC-M subgroup who reported information on LTCs across adulthood the most common sentinel condition was psychological distress, followed by back pain and the most common sequence of accrual of LTCs was the co-reporting of psychological distress and back pain followed by high blood pressure. Just under one-third of the MLTC-M subgroup (30.5%, n=50) reported a variation in the reporting of psychological distress across the adult life course.

### Comparison with other studies
Few studies have described the sequences of accrual of LTCs across the life course or explored the sentinel condition. In terms of the sequence of LTC accrual it is of note that as demonstrated in figure 3, psychological distress on its own or co-reported alongside another condition was a common sentinel condition, and various pathways to multimorbidity emerged from early psychological distress, this is worthy of further exploration in larger datasets with additional LTCs. Our findings support research among an urban multiethnic borough in London, which found that multimorbidity in those under 65 years of age tended to start with depression or serious mental illness.[7] Further, among younger cohorts (age 20–39), the most common sequence of accrual of LTCs began with depression.[38] Our findings are comparable with other studies that have reported the high prevalence of mental ill-health among multimorbidity cohorts.[3 39] There is therefore increasing evidence emphasising the need to target prevention efforts aimed at mental ill-health in the early life course. Future research could consider what role targeted mental health prevention may play in altering multimorbidity pathways.

Research using longitudinal analysis to report sequencing and clusters of LTCs across the life course is scarce.[4] An Australian longitudinal study found that in a cohort of women over the age of 50, 17% developed multimorbidity over a 20-year period.[40] The onset of stroke was most strongly associated with the progression to multimorbidity (23%), whereas a smaller proportion of women progressed to multimorbidity after an initial diagnosis of diabetes (10%) or heart disease (11%).[40] Our research supports a similar life course trajectory timeline: 22% of the cohort took 16 years to accrue all three LTCs and 20% took 20 years. A longitudinal study from Australia considered eight LTCs over a 10-year period; onset of hypertension, cardiovascular disease and chronic obstructive pulmonary disease (COPD) were likely to occur at an older age and hypercholesterolaemia, asthma

**Table 2** Socio-demographic characteristics of MLTC-M group (psychological distress, high blood pressure and back pain), compared with remainder of BCS70 sample at age 46

| | MLTC-M | Non-MLTC-M | Mann-Whitney U tests |
|---|---|---|---|
| | N (%) | N (%) | |
| Sex | | | P=0.737 |
| Male | 173 (46.8) | 3953 (48.4) | |
| Female | 197 (53.2) | 4213 (51.6) | |
| IMD quintiles | | | **P<0.001** |
| 1 (most deprived) | 90 (24.3) | 1039 (12.7) | |
| 2 | 74 (20.0) | 1315 (16.1) | |
| 3 | 87 (23.5) | 1701 (20.8) | |
| 4 | 68 (18.4) | 1981 (24.6) | |
| 5 (least deprived) | 51 (13.8) | 2130 (26.1) | |
| Highest academic qualifications | | | **P<0.001** |
| No qualification | 142 (38.4) | 2263 (27.7) | |
| GCSE or equivalent | 139 (37.6) | 2651 (32.5) | |
| A level or equivalent | 42 (11.3) | 1130 (13.9) | |
| Degree or higher | 47 (12.7) | 2122 (26.0) | |
| Housing tenure | | | **P<0.001** |
| Own/mortgage | 218 (58.9) | 6551 (80.2) | |
| Social rent | 101 (27.3) | 812 (9.9) | |
| Private rent/other | 51 (13.8) | 803 (9.8) | |
| Local authority/housing association | | | **P<0.001** |
| No | 269 (72.7) | 7354 (90.1) | |
| Yes | 101 (27.3) | 812 (9.9) | |
| Employment status | | | |
| Employed | 245 (66.2) | 7239 (88.7) | **P<0.001** |
| Unemployed | 9 (2.4) | 144 (1.8) | |
| Sickness/disability | 72 (19.5) | 271 (3.5) | |
| Caring for family | 35 (9.5) | 410 (5.0) | |
| Other | 9 (2.4) | 102 (1.3) | |
| Benefits (including child benefits) | | | **P<0.001** |
| No | 170 (46.0) | 5171 (63.3) | |
| Yes | 200 (54.0) | 2995 (36.7) | |
| Total | 370 (100.0) | 8166 (100.0) | |

Significant difference was found using a Mann-Whitney U test included in bold—excluding missing.
BCS70, 1970 British Cohort Study ; IMD, Index of Multiple Deprivation ; MLTC-M, multiple long-term condition multimorbidity .

and mental health problem at younger ages.[41] Although hypercholesterolaemia was more likely to occur as a first chronic disease (CD), it was significantly associated with a lower risk of developing further CDs.[26] Alternatively, mood and anxiety disorders or asthma at baseline was associated with the risk of developing further CDs.[41] Those reporting mood and anxiety disorders were at a higher subsequent risk of diabetes, asthma and mental disorders; those reporting asthma were at a higher risk of developing COPD and hypercholesterolaemia.[41]

Finally, a study from the USA explored the acquisition sequence of 20 LTCs describing both two-condition and three-condition multimorbidity.[38] The most common sequence of three-condition multimorbidity in 20–39 years old was depression, asthma and substance misuse, whereas in 50–59 years old, it was hyperlipidaemia, hypertension and diabetes.[38] Our research supports a similar sequence of accrual timeline to previous studies.[38 41] We found psychological distress was likely to be the sentinel condition at younger ages. High blood pressure was infrequently reported as the sentinel condition and likely to be reported at an older age, often as the last condition.

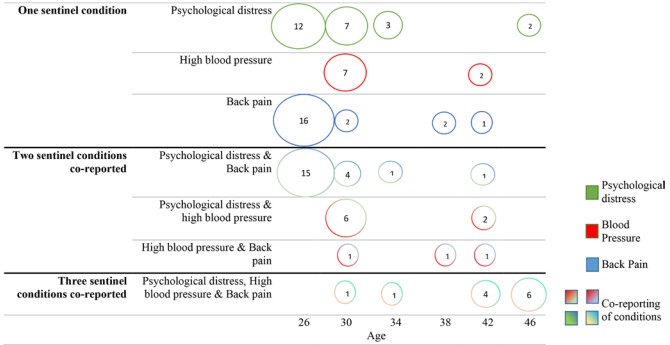

**Figure 2** Bubble diagram of the percentage distribution of sentinel conditions and the age at which the sentinel condition was reported, for all 164 cohort members who reported the multiple long-term condition multimorbidity (MLTC-M) group of psychological distress, high blood pressure and back pain at age 46 and provided information on long-term condition across the adult sweeps. The area of each bubble represents, for those that reported the MLTC-M group of psychological distress, high blood pressure and back pain at age 46, the reporting of the sentinel condition at each sweep in adulthood. The percentages are included inside each bubble.

### Permanency of the psychological distress over time

Previous research suggest that mental health trajectories have been found to influence future health outcomes.[13–16] Persistent psychological distress across sweeps was found in 27.4% (n=45) of the MLTC-M subsample and psychological distress that fluctuated was reported in 30.5% (n=50). This represents a substantial proportion of people with transient psychological distress pathways and supports the need to consider how the cross-sectional reporting of LTC may not reflect the variation in certain LTC over the life course. We demonstrate that the life course pathways to the three-condition MLTC-M we considered fluctuated across the population, and use of cross-sectional methods would not have identified these diverse pathways to multimorbidity. Our results are also distinct from studies using

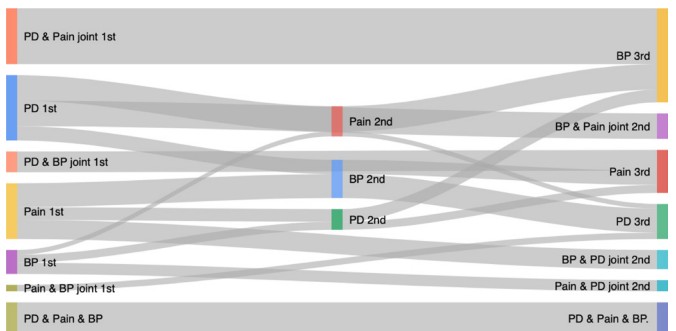

**Figure 3** Sankey diagram of sequences of long-term condition for all 164 cohort members who reported the multiple long-term condition multimorbidity group of psychological distress (PD), high blood pressure (BP) and back pain (pain) at age 46 and provided information on long-term condition across the adult sweeps. The size of the bar represents the number of people within each pathway. Larger bars representing a greater number of people.

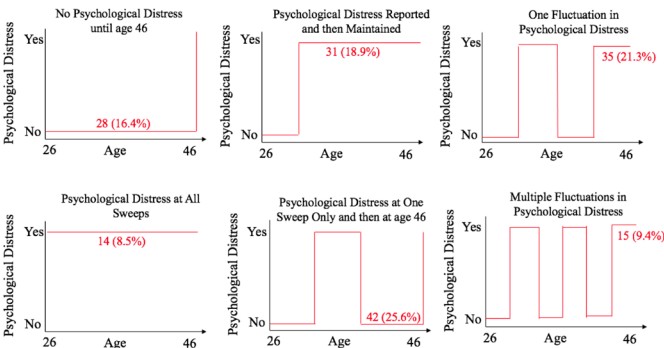

**Figure 4** A graph representing the reporting of psychological distress across the adult sweeps for all 164 cohort members who reported the multiple long-term condition multimorbidity group of psychological distress, high blood pressure and back pain at age 46 and provided information on long-term condition across the adult sweeps.

clinical coding in routine data, where the presence of a diagnosis code often leads to a suggestion of permanency of a diagnosis within healthcare records. Using a validated tool for assessing psychological distress as we have in this study, rather than a purely self-reported measure, demonstrates the greater complexity of trajectories than clinical coding. For example, over 55% (n=93) of the MLTC-M subsample report a psychological distress trajectory that changed at least once across the adult life course. Previous research that has considered psychological distress trajectories found that the BCS70 cohort experienced elevated levels of psychological distress in their mid-20s compared with their early mid-30s,[26] and that midlife appeared to be a particularly vulnerable phase for experiencing psychological distress.[26] Our MLTC-M subgroup showed a similar psychological distress life course trajectory. It is also noteworthy that while we focus on the permanency of psychological distress, other studies have explored the permanency of physical health conditions including asthma,[42] pain[43] and obesity,[44] and this is worthy of further consideration in future research.

### Meaning of the study

Among the BCS70 cohort two-condition multimorbidity at age 46–48 was common (34%).[19] Furthermore, almost one-third of the people in a representative primary care data set who reported two LTCs were younger than 65 years.[3] Ruel et al reported a two-condition multimorbidity prevalence of 32% with a mean age of 50 years.[41] Sauver et al concluded that the total number of people who developed multimorbidity before age 65 years was more than four times greater than the number at ages 65 and over.[38] Our research determined that the most common three-condition MLTC-M in the BCS70 was infrequent (4.3%). However, this still represents a large number of people living with early-onset MLTC-M, and considering all 10 three-condition MLTC-M we find 18.6% of the BCS70 affected. Therefore increasingly conclusive evidence emphasises the need to target multimorbidity prevention efforts at a younger age. This is supported by our findings

that 45.1% (n=74) of the MLTC-M subgroup reported the sentinel condition by age 26.

Multimorbidity prevalence differs according to demographic factors. Barnett *et al* found that the most deprived areas had the same prevalence of multimorbidity as people aged 10–15 years older living in the most affluent areas.[3] Ashworth *et al* concluded that ethnicity and social deprivation were significant determinants of multimorbidity.[7] Sauver *et al* suggested that the incidence of both two-condition and three-condition multimorbidity varied by age and ethnicity.[38] Gondek *et al* also identified socioeconomic circumstances as early life determinants of multimorbidity among the BCS70 cohort.[17] We demonstrate the reporting of the most common three-condition MLTC-M was patterned by socio-demographic factors encompassing both area measures (IMD) and individual-level measures of deprivation (employment, education and benefits).

From a public health perspective, describing the sequence of accrual of LTCs, identifying the sentinel condition and exploring the permanency of the reporting of certain LTCs could provide the opportunity to identify critical time points for population-level prevention efforts to reduce multimorbidity burden. We plan future work to explore the way in which determinants of multimorbidity may be similar or different from determinants of the sentinel condition and sequence of accrual of LTCs on the trajectory towards multimorbidity. Multimorbidity at midlife is potentially more burdensome, particularly for those of working age. In a cross-sectional study of about 800 people with multimorbidity in the UK, higher treatment burden was associated with people aged 55–64 versus those aged over 65.[45] The sentinel condition is also an indicator of a pathway to future health outcomes and therefore identifying the sentinel condition of multimorbidity is important for targeted preventions with the aim of altering pathways to further health outcomes. This paper has demonstrated the potential of using birth cohort data, with repeated measures, to study MLTC-M across the life course. Future work may look to sequence LTCs and the sentinel condition on a larger scale and we hope this paper can be the foundation for this research.

### Strengths and limitations

The use of a prospective longitudinal approach provides the opportunity to analyse the time-sequenced nature of the data.[46] Furthermore, there are multiple wider determinant exposures captured in BCS70, unavailable in most other data sets which will provide opportunities for future life course investigation. Knowing the temporal sequencing of LTCs offers valuable means for elucidating an understanding of the accrual sequence of LTCs. However, the BCS70, like other major cohort studies, is affected by significant attrition and wave non-response. This is a limitation of our study—there were few people who reported both the most common three-condition MLTC-M at age 46 and information on LTCs across all the adult sweeps (164/370). We therefore risk bias from

under-reporting of the MLTC-M group. Further, the small sample size also precludes the opportunity to explore men and women independently. However, this paper is exploratory: we do not conduct any regression analysis and are cautious not to overstate our findings. We recommend that future work using formal statistical methods consider accounting for attrition and wave non-response.

Data quality remains an important limitation. The variables used to assess the sequence of accrual of LTCs did not match up identically across sweeps and we were required to make assumptions about whether variables captured the same LTCs, and this may have accounted for some of the fluctuation in the reporting of psychological distress we observed. Further, at age 26, blood pressure was not recorded, although the number of those reporting high blood pressure by age 26 was likely to be small.[34 35] Data coding constraints meant we were unable to provide a more precise measurement of when the sentinel condition was reported or differentiate between the times of onset of conditions if they were first recorded at the same sweep. Additionally, these data constraints meant our sequencing results were only granular down to the 4-year level. Further work in birth cohorts to harmonise and agree LTC definitions would strengthen their potential for multimorbidity research.

Some LTCs were self-reported and are therefore likely under-recorded, implying that we may be underestimating the true prevalence of multimorbidity. A further limitation of the data is that there may be inconsistency in the reporting of conditions across the sweeps of data collection. Additionally, the cohort is representative of a cohort of children born in 1970, and as such does not reflect the ethnically diverse population of the UK today (online supplemental table 1). It is known from previous research that the prevalence of multimorbidity is more common and occurs earlier in certain ethnic minority populations.[7 38] We are also aware that health is not static and we aim to incorporate further methods capturing fluctuation in other LTCs when exploring the sequence of accrual of LTCs. Further, we focus on the relative risk of psychological distress; this measure is limited to symptoms of depression and anxiety, and does not capture the full range of other mental health problems. In addition, we define burdensomeness as psychological distress occurring as part of multimorbidity. However, there are other causes of burdensomeness such as physical symptoms and factors influencing treatment burden (number of healthcare appointments, polypharmacy and quality of life measures) we did not consider; these are described within the cumulative complexity model,[47] and burden treatment theory.[48] These should be investigated in future work.

Finally, these analyses were exploratory and descriptive. Caution must be taken not to overstate our findings given the limitations of sample size, attrition and the fluctuation in the recording of LTCs. Limitations of these analyses precluded further statistical analysis. Future work should consider multiple wider determinant exposures

captured in birth cohort data sets and explore the associations between the early life course and both sentinel conditions and early-onset burdensome multimorbidity clusters. However, more careful consideration around the role of confounders, mediators, attrition, sample size and the reporting of LTCs are required.

## CONCLUSIONS

In these exploratory analyses in a national birth cohort, while overall prevalence of the most common three-condition burdensome multimorbidity group was relatively low (4.3%) at age 46, we identified psychological distress and back pain as the most common sentinel conditions. The MLTC-M group was characterised by more disadvantaged socio-demographic factors. Birth cohorts may usefully inform life course-focused multimorbidity research but there are current limitations to their utility. We cannot influence the BCS70 sample size and diversity but future work could combine with other birth cohorts that include people into their 60s. Further work is also needed to agree on LTC definitions to strengthen their potential for multimorbidity research.

**Author affiliations**
[1]School of Primary Care, Population Sciences and Medical Education, University of Southampton, Southampton, UK
[2]MRC Lifecourse Epidemiology Centre, University of Southampton Faculty of Medicine, Southampton, UK
[3]Applied Research Collaboration Wessex, NIHR, Southampton, UK
[4]School of Mathematical Sciences, University of Southampton, Southampton, UK
[5]IT Innovation Centre, University of Southampton, Southampton, UK
[6]NIHR Southampton Biomedical Research Centre, University of Southampton, Southampton, UK

**Contributors** SS, EH, NAA, MB, RH, ZZ, SC and SF were involved in the conceptualisation of ideas. SF obtained the 1970 British Cohort Study data. SS, EH, NAA and SF designed and planned the analyses. SS and EH performed the statistical analysis. SS performed initial draft of manuscript. SS, EH, NAA, SC, RH, MB, WW, JM, MA and SF were involved in drafting, editing and reviewing the manuscript. All authors contributed to the development of the project and were involved in the approval of the final manuscript. SS and SF take responsibility for the data and research governance. SS is acting as the guarantor of this research.

**Funding** This research is supported by National Institute for Health Research (NIHR) grant number (NIHR202644) as part of a larger research project titled 'Developing a Multidisciplinary Ecosystem to Study Lifecourse Determinants of Complex Mid-life Multimorbidity using Artificial Intelligence (MELD)'. The funder of the study had no role in the study design, data collection, data analysis, data interpretation or the writing of the paper. The views and opinions expressed in this paper are those of the authors and do not necessarily reflect those of the NIHR or the Department of Health and Social Care.

**Competing interests** None declared.

**Patient and public involvement** Patients and/or the public were involved in the design, or conduct, or reporting, or dissemination plans of this research. Refer to the Methods section for further details.

**Patient consent for publication** Not applicable.

**Ethics approval** Ethical approval for the 1970 British Cohort Study (BCS70) study was granted by the National Health Service Research Ethics Committee, and all participants provided fully informed consent. For details on the BCS70 ethics and consent please see: https://cls.ucl.ac.uk/wp-content/uploads/2017/07/BCS70-ethical-review-and-consent-Shepherd-P-November-2012.pdf. Ethics approval for the MELD project which this analysis forms part of was granted by Southampton University Ethics and Research Governance Online (61451) and by

The Proportionate Review Subcommittee of the London - Camden & Kings Cross Research Ethics Committee (21/PR/0023). Participants gave informed consent to participate in the study before taking part.

**Provenance and peer review** Not commissioned; externally peer reviewed.

**Data availability statement** All data relevant to the study are included in the article or uploaded as supplementary information. The 1970 British Cohort Study (BCS70) datasets generated and analysed in the current study are available in the UK Data Service repository: BCS70 https://discover.ukdataservice.ac.uk/series/?sn=200001.

**ORCID iDs**
Sebastian Stannard http://orcid.org/0000-0002-6139-1020
Simon DS Fraser http://orcid.org/0000-0002-4172-4406

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
