## [Reviewer comments · BMJ Open]

ARTICLE DETAILS

TITLE (PROVISIONAL)	Early-onset burdensome multimorbidity: an exploratory analysis of sentinel conditions, condition accrual sequence and duration of three long-term conditions using the 1970 British Cohort Study
AUTHORS	Stannard, Sebastian; Holland, Emilia; Crozier, S; Hoyle, Rebecca; Boniface, Michael; Ahmed, Mazen; McMahon, James; Ware, William; Zlatev, Zlatko; Alwan, Nisreen; Fraser, Simon

VERSION 1 – REVIEW

REVIEWER	Ashworth, Mark King's College London, UK, School of Population Health & Environmental Sciences
REVIEW RETURNED	10-Dec-2021

GENERAL COMMENTS	Comments: 1) Abstract: a small point but in the Objectives, it states: 'to estimate the prevalence of the most common three-condition early-onset multimorbidity (MLTC-M) group at age 46'. In fact the objective was to estimate prevalence of MLTC-M in middle age. It was only on reading the Abstract: Design that it became clear that 'age 46' was the last of the age-related data extractions. In other words, 'at age 46' wasn't the objective. 2) Abstract: Objectives. The Aim is well summarised in the Objectives and includes: '(to) explore the permanency of the reporting of LTCs'. Does this aim to explore permanency only apply to psychological conditions? This is important because the Abstract: Results only refers to the permanency of psychological conditions. It is only when one comes to the final para of the Introduction, main article, does it become clear that 'permanency' is only applied to psychological distress. In MLTC research, some researchers have also explored permanency of all conditions; an obvious example of permanency of a physical condition of relevance is asthma. Might be worth making reference in the main article to the concept that 'permanency' may apply to both physical and mental health conditions, even though the current analysis was confined to psychological distress. 3) Abstract: Results: it would be clearer if % values were reported first throughout since they convey the main message, with numerator/denominator values reported in brackets. 4) Abstract: Conclusion: if the Abstract Word Count permits, would be good if the e most common three-condition MLTC-M group could be summarised. After all, defining this group was one of the main Aims, as summarised in the Objectives section.
--

	5) Methods: Variables. This section contains some Results (Table 1 results). It would be better to disentangle Results from Methods. Thus, the Methods should best describe that approach to identifying the most common three-condition MLTC-M group and what Methods to apply to this group, once identified. Rather than declaring which three-condition group was identified in advance of describing the Results. 6) Methods: Malaise Index: a bit more detail about the Malaise Index would be of help. The title of this Index is rather odd, but probably reflects practice from several decades ago. The implication is that it is an Index of all forms of malaise, including fatigue, general wellbeing, feeling unwell, etc. In fact, it was designed specifically to measure signs of psychological distress in adulthood (that should be clarified). I am unaware of whether it has been independently validated or only used in the BCS70 (perhaps the authors can clarify). 7) Results, para 2. The authors state: 'a larger proportion of participants (24.3%) lived in areas in the lowest IMD quintile compared with the remainder of the BCS70 sample'. This summary should be accompanied by a P value since the reader will not know whether 'larger' means a mere higher % value or a statistically different, higher % value. The P value should be written $P < 0.001$, and not as $P = 0.000$ as the authors have written in Table 2. P values of zero are 'impossible'. 8) Results, pg11. There seems to be some text missing in the PDF file, as this page begins with a new sentence stating: 'receive benefits (54.0% vs 36.7%) and were more likely to be off work for issues relating to sickness (19.5% vs 3.5%).' At this point the authors do refer to a P value. 9) Results pg10: as per the Abstract, it is easier for the reader to view % values and then the numerator and denominator in brackets thereafter 10) Discussion pg12: para 2 might have meant to state Reference 7 rather than Reference 4, here: 'research among an urban multi-ethnic borough in London, which found that multimorbidity in those under 65 tended to start with depression or serious mental illness.[4]' 11) Discussion pp11/12: the para discussing 'Permanency of the psychological distress over time', could be expanded. I considered that Figure 4 was unique data, showing the patterns of psychological distress (in those who had psychological distress at the final data collection). The Results section simply described the sequence in terms of Yes-No-Yes-Yes etc. This was hard to understand. A more narrative description of patterns, especially the most common one of two patterns would be really helpful and would clarify, emphasising the strengths of this study. 12) Discussion: meaning of the study. The authors report the overall prevalence of 2-LTC multimorbidity, 'at age 46-48 was common (34%)', but don't offer the comparable figure for 3-LTC multimorbidity (which was, after all, a main aim of the study), i.e. the combined prevalence for all ten three-condition MLTC-Ms.
--	---

	13) Discussion: Limitations. The authors should add that studies of LTC sequencing may be hampered by 4-yearly data sweeps, since LTCs acquired during a 4 year data collection period would have appeared simultaneous in the dataset; in other words, sequencing was only granular down to the 4-year level. Also, the study aimed to define 'burdensomeness'; at least that is what the Title implies even though the Introduction narrows this down to psychological distress. This is an important concept and poorly recorded in primary care records although BCS70 data provides the opportunity to study this aspect of burdensomeness. Given that the title uses the term 'burdensome', the Discussion gives the authors the opportunity to place an analysis of psychological distress within the overall concept of burden (since there are other domains to burden, such as nos. of healthcare appointments, polypharmacy, Quality of Life measures, etc). 14) Finally, there is very little in the Discussion about the excellent Figure 3 Sankey plot. At least some discussion seems merited about the main trends illustrated by this Figure.
--	--

REVIEWER	Canizares, Mayilee University Health Network, Arthritis Program
REVIEW RETURNED	14-Jun-2022

GENERAL COMMENTS	This is a descriptive analysis of the co-occurrence of PD, back pain and HBP. I have the following methodological and analytical questions:  - Given the reduction in sample size, why the study limited the data on participants with 3 conditions? It seems to me that similar analysis could be done if the definition of multimorbidity is defined more broadly. - For the analysis of the sentinel conditions the sample was further reduced by more than a half, due to the authors conducting a complete case analysis. CC analyses tend to give biased estimates. I think the authors should run the analysis with all data available, use imputation methods and compared findings to evaluate the impact of missing data on the results. - Another issue, somewhat related to the previous points, is that often in surveys participants are not consistent in reporting chronic conditions by waves of data collection. This may be important as the selection criteria was very stringent. For example, there may be a situation in which participants reported the 3 conditions in an early wave of data collection but failed to report the 3 conditions at age 46, and therefore they could have been excluded from the analysis. I suggest that the authors comment on the potential impact of this issue on their results. General comments:  - This paper focus on the co-occurrence of three chronic conditions, however the title suggest that the study was about multimorbidity more general. I suggest modifying the title to reflect the actual analysis more accurately. - In several places in the manuscript the authors note that "Using birth cohort data affords the opportunity to explore repeated validated measures of mental health across the lifecourse", this is not accurate as is not birth cohort rather than data on repeated measures what allows look at the lifecourse trajectories. However, I would agree that analysis of data from one birth cohort has the
---

	advantage of comparing lifecourse trajectories of people at the same age.  - Abstract/Conclusion: although the last sentence is true I don't think this is a conclusion from this analysis. - There are extra commas before citations across the manuscript.
--	---

VERSION 1 – AUTHOR RESPONSE

Reviewer 1

1. Abstract: a small point but in the Objectives, it states: 'to estimate the prevalence of the most common three-condition early-onset multimorbidity (MLTC-M) group at age 46'. In fact the objective was to estimate prevalence of MLTC-M in middle age. It was only on reading the Abstract: Design that it became clear that 'age 46' was the last of the age-related data extractions. In other words, 'at age 46' wasn't the objective.

Thank you, we have now changed 'at age 46' to 'at midlife' in the abstract. We have decided to use the term midlife rather than middle age as this is a more frequently used within the literature.

2. Abstract: Objectives. The Aim is well summarised in the Objectives and includes: '(to) explore the permanency of the reporting of LTCs'. Does this aim to explore permanency only apply to psychological conditions? This is important because the Abstract: Results only refers to the permanency of psychological conditions. It is only when one comes to the final para of the Introduction, main article, does it become clear that 'permanency' is only applied to psychological distress. In MLTC research, some researchers have also explored permanency of all conditions; an obvious example of permanency of a physical condition of relevance is asthma. Might be worth making reference in the main article to the concept that 'permanency' may apply to both physical and mental health conditions, even though the current analysis was confined to psychological distress.

Thank you, we have updated the abstract to make it clear to the reader that we are only considering the permanency of psychological distress, and we have added additional information on page 4 to make the reader aware that the reporting of both long term physical and mental health may not be permanent. Additionally, we now note (on page 11) that although we focus on the permanency of psychological distress other studies have focused on the permanency of physical health conditions including asthma (Holgate & Davies, 2009), pain (Dunn et al., 2006) and obesity (Carmen & Cunningham, 2020).

3. Abstract: Results: it would be clearer if % values were reported first throughout since they convey the main message, with numerator/denominator values reported in brackets.

We have now reviewed the manuscript and report the percentage values first with the numerator and denominator now reported in brackets.

4. Abstract: Conclusion: if the Abstract Word Count permits, would be good if the most common three-condition MLTC-M group could be summarised. After all, defining this group was one of the main Aims, as summarised in the Objectives section.

Thank you, we have now added this additional information in the abstract conclusion.

5. Methods: Variables. This section contains some Results (Table 1 results). It would be better to disentangle Results from Methods. Thus, the Methods should best describe that approach to identifying the most common three-condition MLTC-M group and what Methods to apply to this group, once identified. Rather than declaring which three-condition group was identified in advance of describing the Results.

Thank you, given that this paper was descriptive and involved scoping work it was important that we discussed the three-condition group first as this led into the rest of the methods section outlining the variables we then used for the sequencing. However, we have now moved the reference to Table 1 from the methods section to the results section (page 7). We have additionally renamed the heading for this sub-section 'Selecting MLTC-M outcome at age 46' and reconsidered some of the wording. Finally, we now cite some of our published early scoping work on selecting MLTC-M and sequencing (Holland et al., 2021).

6. Methods: Malaise Index: a bit more detail about the Malaise Index would be of help. The title of this Index is rather odd, but probably reflects practice from several decades ago. The implication is that it is an Index of all forms of malaise, including fatigue, general wellbeing, feeling unwell, etc. In fact, it was designed specifically to measure signs of psychological distress in adulthood (that should be clarified). I am unaware of whether it has been independently validated or only used in the BCS70 (perhaps the authors can clarify).

Thank you, these points are well made and we agree that greater clarity will improve the paper. We have therefore now added further explanation about the Malaise index on page 5. We now specifically refer to papers that have found a good level of internal consistency and that have found the index validity to be robust across different populations including the general population and high-risk groups. We also now explain that 'psychological distress' measured via the Malaise Index is a term developed by the UK data service,[24], and used widely within the context of the British birth cohort studies (page 5).

7. Results, para 2. The authors state: 'a larger proportion of participants (24.3%) lived in areas in the lowest IMD quintile compared with the remainder of the BCS70 sample'. This summary should be accompanied by a P value since the reader will not know whether 'larger' means a mere higher % value or a statistically different, higher % value. The P value should be written $P < 0.001$, and not as $P = 0.000$ as the authors have written in Table 2. P values of zero are 'impossible'.

As requested, we now accompany the results with a P value. The P values now state $P < 0.001$ rather than $P = 0.000$.

8. Results, pg11. There seems to be some text missing in the PDF file, as this page begins with a new sentence stating: 'receive benefits (54.0% vs 36.7%) and were more likely to be off work for issues relating to sickness (19.5% vs 3.5%)'. At this point the authors do refer to a P value.

In the previous version on the manuscript the text had been 'split' around a table. This has now been addressed and there is no missing text.

9. Results pg10: as per the Abstract, it is easier for the reader to view % values and then the numerator and denominator in brackets thereafter.

We have reviewed the manuscript and report the percentage values first with the numerator and denominator reported in brackets.

10. Discussion pg12: para 2 might have meant to state Reference 7 rather than Reference 4, here: 'research among an urban multi-ethnic borough in London, which found that multimorbidity in those under 65 tended to start with depression or serious mental illness,[4]'

Thank you for bringing this to our attention, we have changed the reference from 4 to 7. We apologise for this error.

11. Discussion pp11/12: the para discussing 'Permanency of the psychological distress over time', could be expanded. I considered that Figure 4 was unique data, showing the patterns of psychological distress (in those who had psychological distress at the final data collection). The Results section simply described the sequence in terms of Yes-No-Yes-Yes etc. This was hard to understand. A more narrative description of patterns, especially the most common one of two patterns would be really helpful and would clarify, emphasising the strengths of this study.

Thank you for this comment, we have added a more descriptive narrative of the patterns of the permanency of psychological distress on page 9. Further, this helpful comment has stimulated a number of discussions amongst the authorship team and as such we have expanded discussions on the permanency of psychological distress (page 11). We now discuss how our analysis demonstrated that the lifecourse pathways to the three condition MLTC-M we considered fluctuated across the population and using cross sectional methods would not reflect these diverse pathways to multimorbidity. Additionally, we also discuss how our results are distinct from clinical coding in routine data, where clinical coding assumes a permanency of a diagnosis within health care records. Using the validated malaise tool for assessing psychological distress, rather than relying purely on self-report, we have shown that psychological distress trajectories are more complicated than would likely be seen using clinical coding alone. We have now alluded to this in the discussion and are grateful to the reviewer for raising this issue.

12. Discussion: meaning of the study. The authors report the overall prevalence of 2-LTC multimorbidity, 'at age 46-48 was common (34%)', but don't offer the comparable figure for 3-LTC multimorbidity (which was, after all, a main aim of the study), i.e. the combined prevalence for all ten three-condition MLTC-Ms.

Thank you, we now refer to the combined prevalence for all ten three-conditions MLTC-Ms on page 11.

13. Discussion: Limitations. The authors should add that studies of LTC sequencing may be hampered by 4-yearly data sweeps, since LTCs acquired during a 4 year data collection period would have appeared simultaneous in the dataset; in other words, sequencing was only granular down to the 4-year level. Also, the study aimed to define 'burdensomeness'; at least that is what the Title implies even though the Introduction narrows this down to psychological distress. This is an important concept and poorly recorded in primary care records although BCS70 data provides the opportunity to study this aspect of burdensomeness. Given that the title uses the term 'burdensome', the Discussion gives the authors the opportunity to place an analysis of psychological distress within the overall concept of burden (since there are other domains to burden, such as nos. of healthcare appointments, polypharmacy, Quality of Life measures, etc).

We have added to our limitation section on page 12. We now outline that, as suggested, our sequencing is only granular down to the 4 year level. We also now discuss that physiological distress is only one aspect within the wider concept of burden (other factors include physical health and treatment burden).

14. Finally, there is very little in the Discussion about the excellent Figure 3 Sankey plot. At least some discussion seems merited about the main trends illustrated by this Figure.

We agree this is a good suggestion and we have added some additional discussion about the Sankey plot on page 11 (under the subheading 'comparison with other studies').

Reviewer 2

1. Given the reduction in sample size, why the study limited the data on participants with 3 conditions? It seems to me that similar analysis could be done if the definition of multimorbidity is defined more broadly.

Thank you, we chose to study three conditions for several reasons. Firstly, the paper was exploratory: we wanted to understand and explore the patterns and sequence of multimorbidity alluding to the role of the sentinel condition. Evaluating three conditions allowed us to achieve these aims in more detail than if we were to have considered two conditions (even though two conditions is often considered sufficient for a definition of multimorbidity). Additionally, two condition multimorbidity has been defined

in the BCS70 at age 46 by Gondek et al., (2021), and we thought an important extension to this work would be to describe three condition multimorbidity at the same timepoint. Furthermore, given the limitations of birth cohorts datasets we were limited by number of conditions we could explore (compared to primary care data for example), this therefore precluded the opportunity to explore multimorbidity more widely. Instead, we chose to focus on the ten most commonly reported long-term conditions from the data available.

2. For the analysis of the sentinel conditions the sample was further reduced by more than a half, due to the authors conducting a complete case analysis. CC analyses tend to give biased estimates. I think the authors should run the analysis with all data available, use imputation methods and compared findings to evaluate the impact of missing data on the results.

We understand the reviewer's concern about complete case and the use of multiple imputation. However, we do not think it would be appropriate for this paper. We did not conduct any analysis that included an estimate of effect size, for example univariable or multivariable regression (e.g. an exposure-outcome relationship); this was deliberate as our paper was only descriptive and exploratory. We intend to make use of multiple imputation for our currently funded work using three birth cohorts; but for the present analyses where it was not the aim to find definitive effect sizes but rather initial scoping and descriptive work, we prefer to present the complete-case analyses.

3. Another issue, somewhat related to the previous points, is that often in surveys participants are not consistent in reporting chronic conditions by waves of data collection. This may be important as the selection criteria was very stringent. For example, there may be a situation in which participants reported the 3 conditions in an early wave of data collection but failed to report the 3 conditions at age 46, and therefore they could have been excluded from the analysis. I suggest that the authors comment on the potential impact of this issue on their results.

We agree with the reviewer and were limited by the word count to go into detail. We acknowledge that a limitation of self-report data is that there may be inconsistency in the reporting of conditions across the sweeps of data collection and we now acknowledge this limitation on page 12. We also now state that the variables used to assess the sequence of accrual of LTCs do not match up identically across sweeps, and this may have accounted for some of the fluctuation in psychological distress that we observed (page 12). We should state however that psychological distress is based on a range of questions that form a validated tool for estimating psychological distress rather than self-reported measures, and this should help to reduce some measurement error. Finally, it is worth noting that given this paper was exploratory and we deliberately restricted our sample to focus the paper on

cohort members who reported the three conditions MLTC-M at age 46. However, those that did not report the 3 conditions at this age were not excluded from the analysis but instead included in the denominator for the purpose of reporting overall prevalence. Therefore we are likely to be presenting conservative estimates of multimorbidity prevalence.

4. This paper focus on the co-occurrence of three chronic conditions, however the title suggest that the study was about multimorbidity more general. I suggest modifying the title to reflect the actual analysis more accurately.

Thank you, we have now changed the title to: Early-onset burdensome multimorbidity: an exploratory analysis of sentinel conditions, condition accrual sequence and duration of three long-term conditions using the 1970 British Cohort Study.

5. In several places in the manuscript the authors note that “Using birth cohort data affords the opportunity to explore repeated validated measures of mental health across the lifecourse”, this is not accurate as is not birth cohort rather than data on repeated measures what allows look at the lifecourse trajectories. However, I would agree that analysis of data from one birth cohort has the advantage of comparing lifecourse trajectories of people at the same age.

Thank you, where appropriate in the manuscript we now state ‘ Using birth cohort data, with repeated validated measures, affords the opportunity to explore mental health trajectories across the lifecourse.’

6. Abstract/Conclusion: although the last sentence is true I don't think this is a conclusion from this analysis.

Thank you, we have amended this sentence in the abstract conclusion to ensure we have been more cautious with our concluding sentence.

7. There are extra commas before citations across the manuscript.

Thank you, to confirm this was a requirement of the journal.

References

Carmen D, & Cunningham SA. In, out, and fluctuating: obesity from adolescence to adulthood, *Ann Epidemiol* 2020; 41: 14-20.

Dunn KM, Jordan K, & Croft P. Characterising the course of low back pain: a latent class analysis. *Am J Epidemiol*. 2006; 163: 754-761.

Gondek D, Bann D, Brown M, et al. Prevalence and early-life determinants of mid-life multimorbidity: evidence from the 1970 British birth cohort. *BMC Public Health* 2021, 1319.

Holgate ST, & Davies DE. Rethinking the Pathogenesis of Asthma, *Immunity* 2009; 31(3), 362-367.

Holland E, Stannard S, Alwan N, et al. Exploring sentinel conditions and the accrual sequence of multiple long-term condition multimorbidity using birth cohort and primary care data: an exploratory retrospective cohort study [abstract]. *Lancet* 2021; 398: S54.

May CR, Eton DT, Boehmer K, et al. Rethinking the patient: using Burden of Treatment Theory to understand the changing dynamics of illness. *BMC Health Serv Res* 2014; 14, 281.

Shippee ND, Shah ND, May CR, et al. Cumulative complexity: a functional, patient-centered model of patient complexity can improve research and practice. *J Clin Epidemiol* 2012; 65(10), 1041–1051.

VERSION 2 – REVIEW

REVIEWER	Canizares, Mayilee University Health Network, Arthritis Program
REVIEW RETURNED	14-Sep-2022
GENERAL COMMENTS	I have no further queries.